# Assessing the ecological regime and spatial spillover effects of a reclaimed mining subsided lake: A case study of the Pan'an Lake wetland in Xuzhou

**Jiaxing Xu**[1], **Pengcheng Yin**[2,3], **Wenmin Hu**[1]*, **Lili Fu**[4], **Hua Zhao**[3]

**1** The National and Local Joint Engineering Laboratory of Internet Applied Technology on Mines, China University of Mining and Technology, Xuzhou, China, **2** Natural Resources and Planning Bureau of Xuzhou, Xuzhou, China, **3** School of Environmental Science and Spatial Informatics, China University of Mining and Technology, Xuzhou, China, **4** Jiangsu Zenith Geo-Informatics Co., Ltd, Xuzhou, China

* huwm@cumt.edu.cn

**Data Availability Statement:** SPOT5 image acquired on October 16, 2008 can be inquired through the website (http://eds.ceode.ac.cn) and China's GF2 images acquired on October 19 and

## Abstract

In the North China Plain, coal mining subsided lakes are surface water bodies that form after the conclusion of coal mining. In China, mining subsided lakes are often transformed into artificial wetland parks for ecological restoration. While many studies have focused on evaluating coal mining subsidence lake ecosystem service value and water pollution, little attention has been paid to changes in ecological regimes and ecological spillover effects before and after the reclamation of mining areas. This paper examines the Pan'an Lake artificial wetland in Jiawang District, Xuzhou, as a case study. Changes in the ecological regime of the mining subsidence area before and after land reclamation and corresponding spatial spillover effect on land prices are assessed based on remote sensing, GIS raster calculations and geostatistical methods. The results show that the ecosystem service value and ecological storage capacity changed significantly after the mining subsided lake was transformed into an artificial wetland and the wetland ecosystem has been developing well with significantly positive spillover effects on surrounding land prices. From 2008 to 2017, service functions of the artificial wetland ecosystem of Pan'an Lake increased by 81.95%, and the system's ecological storage capacity increased from RMB 6,754 yuan/hm$^2$ in 2008 to RMB 12,289 yuan/hm$^2$ in 2017. The average impact of the Pan'an Lake artificial wetland on the spillover effects of surrounding residential land prices was measured at RMB 195.18 yuan/m$^2$, and the total spillover value of planned residential land in the study area was measured at RMB 805,422,100 yuan. The present study can serve as a useful guide for evaluating the economic feasibility of land reclamation planning and ecological restoration in mining subsidence areas.

## Introduction

Ecosystems in a mining area are complex and shaped by interactions between the natural environment and the social environment dominated by the local development and utilization of

29, 2017 can be inquired through the website for China Resource Satellite Application Center (http://36.112.130.153:7777/DSSPlatform/productSearch.html). The data of land price samples is within the manuscript and its Supporting Information files.

**Funding:** This work is jointly supported by the National Natural Science Foundation of China (41401610, 41601500 and 51874278) and National Key Research and Development Program of China (2017YFC0804401) and the Independent Research Project of State Key Laboratory of Coal Resources and Safe Mining (SKLCRSM2020X04). The funders had no role in study design, data collection and analysis, decision to publish, or preparation of the manuscript. The commercial company 'Jiangsu Zenith Geo-Informatics Co., Ltd' provided support in the form of salaries for the author Lili Fu, but did not have any additional role in the study design, data collection and analysis, decision to publish, or preparation of the manuscript.

**Competing interests:** The authors have declared that no competing interests exist. The commercial company 'Jiangsu Zenith Geo-Informatics Co., Ltd' does not provide experimental funds and will not use the relevant information of the paper to develop products and apply for patents, and will not mind appearing as an author unit. This does not alter our adherence to PLOS ONE policies on sharing data and materials.

mineral resources [1, 2]. Long-term, large-scale and high-intensity mineral resource development activities have destroyed the geological conditions of native deposits, giving rise to surface subsidence, soil pollution and plant destruction as well as changes in hydrothermal structures and damage to the ecological environment, rendering ecosystems extremely vulnerable [3–6]. The average rate of land collapse per 10,000 tons of coal mining in China has been estimated at 0.20–0.33 $hm^2$ [7]. China's cumulative subsidence area created due to coal resource exploitation has been estimated to cover more than 1.5 million $hm^2$ by 2020 and will continue to increase at a rate of 30,000 to 47,000 $hm^2$ every year [6]. Therefore, the comprehensive treatment of mining subsidence areas has become a major focus of research on the sustainable development of mining areas.

The mining subsidence area in the North China Plain is a unique mining area with high groundwater levels. As local groundwater levels are relatively shallow, the subsidence area can flood easily, forming a large waterlogged area. As the subsided area expands due to coal mining, the permanent ponding area may become a pond or even a lake and namely a coal mining subsidence lake, which may form a wetland ecosystem over the long term [8]. As the balance of the water system is compromised, the water body may be polluted by mine water, leaching water from waste heaps, etc., destroying the landscape and reducing land productivity. Such areas thus have very few land-use effects on agricultural or urban development and are considered economic "scars" [9–12]. As an important stage of mine production and construction, land reclamation serves as an important means to minimize environmental damage caused by mining and to restore local ecological balance [13]. In recent years, several mining subsided lakes in China have been transformed into artificial wetland parks according to local conditions using land reclamation techniques as a means of ecological restoration [14]. The rectification of mining subsidence lake areas is a challenging stage of mining area ecological restoration and resource utilization. The successful rectification of a mining subsidence lake is subject to ecosystem restoration conditions and economic benefits of mining areas after reclamation. Therefore, it is of great practical and theoretical significance to evaluate and research reclamation area ecosystems for the implementation of land reclamation and ecological restoration projects in mining areas.

The land reclamation benefits of mining areas are mostly measured based on economic benefits, such as increases in cultivated land and increases and decreases in built-up land, while less attention is paid to ecological environmental effects after reclamation [12, 15]. On one hand, the ecological restoration of a reclaimed mining area involves a long-term restoration process, and due to the influence of reclamation measures, coal mining and other factors, ecological restoration is often not supported with long-term quantitative monitoring. On the other hand, as public resources, ecological environments have obvious spatial spillover effects [16]. However, current research on the ecological environments of reclaimed mining areas mainly focuses on ecological problems experienced within such areas based on evaluations of ecosystem services, water pollution and nutrition status monitoring, landscape ecological restoration, etc. [17–19] while few studies examine impacts of the ecological restoration of a mining area on the surrounding environment such as spillover effects of the improvement of an ecological environment on surrounding land prices.

In recent years, the ecological environments of reclaimed mining areas have mainly been evaluated using the single index method and comprehensive index method. Sun et al. [12] showed that converting a mining subsided lake into a restored wetland through land reclamation and ecological restoration can provide economic and ecological benefits via the recovery of ecological services and functions. Throughout the mining cycle, Demirbugan [17] analysed the net benefits of ecosystem services and found profile changes during mining and rehabilitation processes in the Soma coal region of western Turkey. Wang et al. [18] proposed an eco-

environmental benefit evaluation method based on land-use changes and evaluated changes in eco-environmental benefits during the reclamation of abandoned land in the Fushun Xilutian Mine. Their results show that three main eco-environmental functions (soil formation and protection, biodiversity protection and air regulation) improved most through reclamation, accounting for roughly half of eco-environment benefits achieved. Land prices are typically evaluated using cost approaches, hypothetical development methods, market comparison methods, etc. [20, 21]. However, due to the distinct regional characteristics, short formation periods, immature land price markets and other features of artificial wetlands of mining subsided lakes, the above methods will generate large deviations and are thus not suitable for use. Based on variable correlations and variability, the Kriging spatial interpolation method performs an unbiased and optimized estimation of the values of regionalized variables for a finite region, which offers certain advantages for data with low densities and uneven distributions [22]. Considering that land prices are expressed as land price sample points in space, a change in regional land prices can be reflected by the spatial interpolation of land price samples.

This paper focused on the Pan'an Lake wetland in Jiawang District, Xuzhou. Ecosystem service value and the ecological storage index were used to analyse ecosystem changes in the mining subsidence lake before and after land reclamation, and spatial spillover effects were evaluated via GIS raster calculations and geostatistics. The results provide technical support for assessing land reclamation and ecological restoration effects in mining areas and identify scientific methods for determining reclamation models for mining areas.

## Materials and methods

### Study area

The Pan'an Lake wetland is an artificial wetland produced from the restoration of a mining subsided lake in the southwest of Jiawang District, Xuzhou (117˚19'57"-117˚26'15"E, 34˚ 19'17"-34˚23'33"N). Due to resource exhaustion, coal mining subsidence, ecological degradation and so on, levels of economic development in the Jiawang Mining Area, where the Pan'an Lake wetland is located, are far lower than those of the main urban area of Xuzhou. Since 2010, Jiawang District has implemented land reclamation and ecological restoration projects in the mining subsidence area. As China's first wetland park designed for the ecological restoration of a coal mining subsidence area, the Pan'an Lake wetland was constructed based on a four-in-one model integrating "basic farmland rectification, coal mining subsidence reclamation, ecological environment restoration and wetland landscape development". A national 4A-level ecological wetland park with a water area of 9.21 km$^2$ and an overall area of 52.89 km$^2$ has been constructed. The location of the study area is shown in Fig 1.

### Data sources and processing

The data used in this work were mainly drawn from the multi-temporal remote sensing images and land price survey data for 2008–2017. Table 1 shows the images including SPOT5 and China's GF2 images used in this work. SPOT5 image acquired on October 16, 2008 can be inquired through the website (http://eds.ceode.ac.cn) and China's GF2 images acquired on October 19 and 29, 2017 can be inquired through the website for China Resource Satellite Application Center (http://36.112.130.153:7777/DSSPlatform/productSearch.html). First, three remote sensing images were accurately registered using ERDAS software. Two GF2 images were processed by images mosaic. Random Forest (RF) classification was combined with visual interpretation to obtain land-use data for before and after the reclamation of the study area. Based on characteristics of the study area, landscapes in the study area were divided into 7 types: cultivated land, woodland, grassland, transportation land, natural water bodies

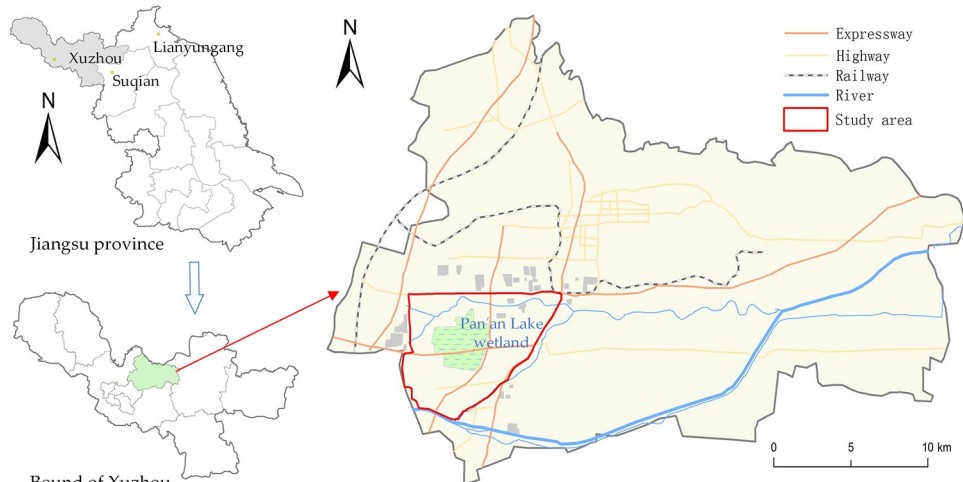

**Fig 1. Location of the study area in Jiawang District, Xuzhou City, Jiangsu Province, China.** The two index maps on the left were created using ArcGIS software and the map on the right was drawn using ArcGIS software with OpenStreetMap from USGS National Map viewer without copyright restrictions. The study area was highlighted in red.

(including rivers and ponds), subsided water bodies, and built-up land (including urban/rural and industrial/mining built-up land). All Kappa coefficients exceed 0.95, meeting the data accuracy requirements of this study. Land price samples data was from Jiawang land price dynamic monitoring sample survey, which were the result of our long-term monitoring and evaluation of land prices. The base date of evaluation of the land price samples were 31 December 2008 and 31 December 2017, respectively. Price samples were collected in the study area for 47 residential lands in 2008 and 65 residential lands in 2017, for which spatial interpolation was performed using GIS and geostatistical methods.

## Methodology

Land reclamation causes changes in landscape patterns and ecological processes by changing the ecological environments in mining areas through the implementation of various reclamation measures, ultimately affecting the provision of ecosystem services and the surrounding environment [18, 23]. To relate abstract ecological concepts to specific economic benefits to better evaluate and measure ecological restoration conditions in the study area, we conducted analyses focused on three variables: ecosystem service value, ecological storage capacity and spillover effects.

**Ecosystem service value evaluation model.** The ecosystem service value evaluation model proposed by Costanza et al. [24] was adopted in this study.

$$ESV = \sum A_i \times VC_i \tag{1}$$

**Table 1. Description of imagery data used for land-use classes in the study area.**

| Imagery data | Imagery type | Resolution | Path and row |
|:---:|:---:|:---:|:---:|
| 10/16/2008 | SPOT 5 | 2.5 m | 286/281 |
| 10/19/2017 | GF2 | 0.8 m | 1012/154 |
| 10/29/2017 | GF2 | 0.8 m | 1013/154 |

**Table 2. Comparison of ecosystem service values per unit of area for the natural and subsided water bodies.**

| Ecosystem Service | Natural Water Body | | Subsided Water Body | |
|---|---|---|---|---|
| | Ecosystem Service | Value (yuan/hm$^2$) | Ecosystem Service | Value (yuan/hm$^2$) |
| Climate regulation | √ | 407 | √ | 407 |
| Water conservation | √ | 18033.2 | √ | 4508.3 |
| Soil formation and conservation | √ | 8.8 | × | 0 |
| Waste treatment | √ | 16086.6 | √ | 4021.65 |
| Biodiversity maintenance | √ | 2203.3 | √ | 1101.65 |
| Food production | √ | 88.5 | √ | 88.5 |
| Raw material | √ | 8.8 | √ | 8.8 |
| Recreation and culture | √ | 3840.2 | × | 0 |
| Total | | 40676.4 | | 10135.9 |

where *ESV* is ecosystem service value (RMB), $A_i$ is the area of Type *i* land use (hm$^2$) and $VC_i$ is the ecosystem service value coefficient of Type *i* land use (RMB/hm$^2$·a). As a unique form of surface water in the mining subsidence area, the subsided water body has caused the mining area to evolve from a single farmland terrestrial ecosystem into a wetland / aquatic ecosystem with climatic regulation, water conservation and raw material supply functions. Meanwhile, the mining area is susceptible to various forms of pollution, including "industrial waste", pesticides, fertilizers, domestic sewage and other pollutants and performs limited soil formation and protection, entertainment and cultural functions. Ecosystem service value coefficients of natural and collapsed water per unit of area are shown in Table 2 [3]. The service value system per unit of area for China's ecosystems formulated by Xie et al. [25] and the value systems of mining cities, towns and industrial ecosystems estimated by Zhang et al. [26] are used, and the value coefficient of ecological services for the study area is obtained (Table 3).

**Ecological storage model.** The ecological storage state (ESS) describes the correlation between land-use changes and their ecological responses. It is a comprehensive expression of ecological changes shaped by land-use quantity, quality, type and distribution determined by possible past and future natural or human activities. The ecological storage state reflects levels of comprehensive ecological storage in a region and is determined by one or more dominant land-use types in a region. The formula is as follows [27]:

$$ESS = \frac{1}{h} \cdot \frac{\sum_{i=1}^{n} ESV_i}{\sum_{i=1}^{n} A_i} \tag{2}$$

where *ESS* is the ecological storage state in the mining area, *h* is the estimated number of years corresponding to the ecological storage state in the mining area (usually, *h* = 1 for the ecological storage state of a certain year), $ESV_i$ is the ecological service value corresponding to ecosystem Type *i*, and $A_i$ is the area of ecosystem Type *i*.

**Spillover effect measurement model.** Land prices are affected by various factors. First, we analysed the influencing factors of land prices in the study area. Affected by housing price

**Table 3. Value coefficients of service functions of different terrestrial ecosystems in the Pan'an Lake wetland.**

| Ecosystem Type | Cultivated land | Wood-land | Grass-land | Natural water body | Subsided water body | Industry/transportation | Urban |
|---|---|---|---|---|---|---|---|
| Value Coefficients | 6114.3* | 19334* | 6406.5* | 40676.4* | 10135.9* | -5372.1* | -1760.5* |

*Unit: yuan·hm$^{-2}$·a$^{-1}$

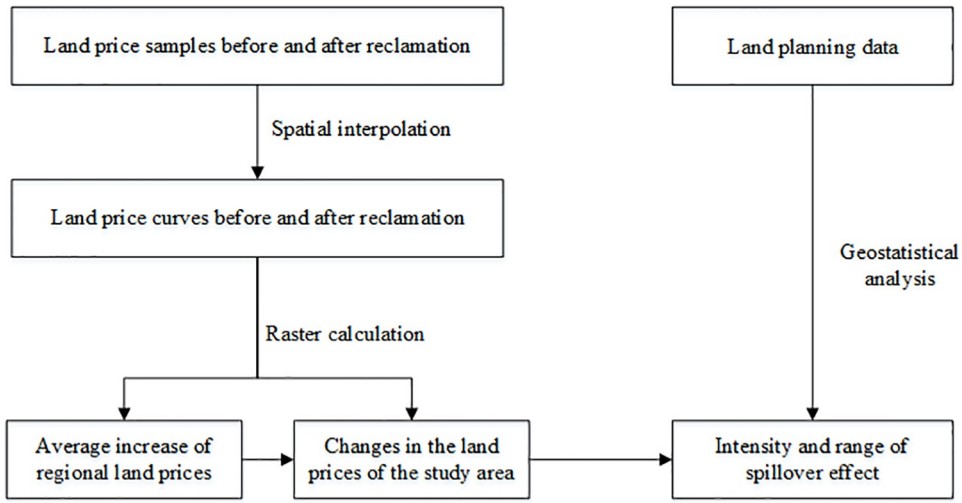

**Fig 2. Spillover effect model framework of land prices for the Pan'an Lake wetland.**

conditions in China, housing and land prices in Jiawang District are increasing rapidly. In addition, besides the ecological restoration project, no other major projects and factors currently affect land prices in the study area. Therefore, impacts of ecological restoration on land prices can be considered as long as the average increase in the value of land prices in the region is removed [28]. Spillover effects were calculated as follows (Fig 2).

1. Calculate the average increase in the value of land prices in the subsided area during the study period and analyse general changes in regional land prices.

2. Calculate the increase in the value of land prices around the study area during the study period and analyse the change in surrounding land prices.

3. Increases in land prices above the average increase in land prices are taken as the impact intensity and range of spillover effects of the reclaimed mining subsided area on surrounding land prices.

Based on the above principles, we propose the following equation for calculating spillover effects of the subsided lake wetland on surrounding land prices:

$$SEV = \sum (Lp_{ij}^{\alpha} - Lp_{ij}^{\beta} - Lp_{ij}^{\beta} \cdot r_i) \times S_{ij} \tag{3}$$

where $SEV$ is the value of spillover effects of the subsided lake wetland on surrounding land values based on the increase in the value of land prices. $Lp_{ij}^{\alpha}$ and $Lp_{ij}^{\beta}$ are the land prices of Plot $j$ of Type $i$ land after and before land reclamation, respectively; $r_i$ is the average rise in Type $i$ land prices in the area of the subsided lake wetland; and $S_{ij}$ is the area of Plot $j$ of Type $i$ land in the subsided area.

## Results and analysis

### Changes in land use and ecosystem service value

According to interpretation results of remote sensing images for 2008 and 2017, landscape types and their changes before and after reclamation in the study area are shown in Fig 3 and

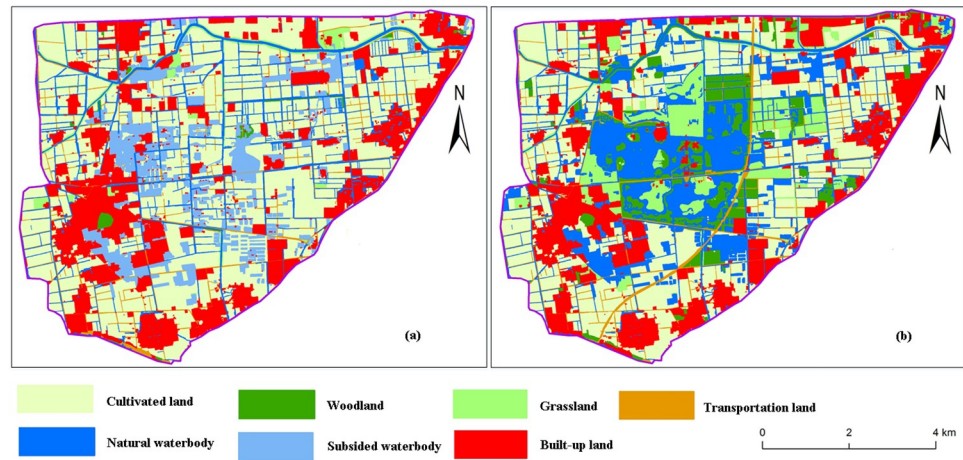

**Fig 3. Landscape patterns before and after land reclamation in the Pan'an Lake subsidence area.** (a) Landscape types in 2008; (b) Landscape types in 2017. The figures are based on the results of remote sensing image interpretations conducted using ENVI software.

Table 4. As can be seen from Table 4, there are 7 types of land use in the study area, among which cultivated land is the prominent. In 2008, cultivated land accounted for more than half of the total area. With the rehabilitation of the ecological environment of the Pan'an Lake wetland park, the proportion of cultivated land decreased to 39.27% from the original value of 54.98%. Next, proportions of woodland and grassland increased from 0.63% and 2.43% to 8.47% and 9.39%, respectively. Subsided water bodies were transformed into lake wetlands through land reclamation, and the area expanded by 16.55% to 152.15 hm$^2$. With the exhaustion of coal resources and reclamation of subsidence areas in Jiawang District, the industrial and mining built-up land area decreased by 12.14% due to the closure of coal mines, the relocation of villages and the gradual withdrawal of abandoned mining land.

As shown in Table 5, the value of ecosystem services in the study area increased by 81.95%. Among them, the value of individual ecological services from forest land, grassland and water bodies increased by 1238.05%, 287.28% and 218.38%, respectively, mainly because coal mining subsidence land was reclaimed into large areas of artificial wetland, woodland, grassland, etc. during mining area land reclamation, causing ecosystem service values to increase. The ecosystem service values of urban land, industrial land, cultivated land, and subsided water bodies decreased mainly due to artificial wetland projects, the removal of coal mining villages and the

**Table 4. Landscape structure changes in the Pan'an Lake subsidence area before and after reclamation.**

| Landscape type | Before land reclamation | | After land reclamation | |
|---|---|---|---|---|
| | Area/hm$^2$ | Percentage /% | Area/hm$^2$ | Percentage /% |
| Cultivated land | 2918.97 | 54.98 | 2084.89 | 39.27 |
| Woodland | 33.62 | 0.63 | 449.84 | 8.47 |
| Grassland | 128.75 | 2.42 | 498.67 | 9.39 |
| Transportation land | 209.69 | 3.95 | 223.85 | 4.22 |
| Natural water body | 336.59 | 6.34 | 1071.63 | 20.18 |
| Built-up land | 1098.91 | 20.70 | 980.54 | 18.47 |
| Subsided water body | 582.89 | 10.98 | 0.00 | 0.00 |
| Total | 5309.42 | 100 | 5309.42 | 100 |

**Table 5. Changes of ESV and value structures for the Pan'an Lake subsidence area before and after reclamation.**

| Landscape Type | Before land reclamation | | After land reclamation | | Changes in ESV | |
|---|---|---|---|---|---|---|
| | ESV | Contribution ratio(%) | ESV | Contribution ratio(%) | ESV | Change rate(%) |
| Cultivated land | 1784.75 | 49.77 | 1274.77 | 19.54 | -509.98 | -28.57 |
| Woodland | 65 | 1.81 | 869.73 | 13.33 | 804.73 | 1238.05 |
| Grassland | 82.49 | 2.3 | 319.47 | 4.9 | 236.98 | 287.28 |
| Transportation land | -112.65 | -3.14 | -128.31 | -1.97 | -15.66 | 13.90 |
| Natural water body | 1369.12 | 38.18 | 4359.02 | 66.81 | 2989.9 | 218.38 |
| Built-up land | -193.46 | -5.39 | -169.98 | -2.61 | 23.48 | -12.14 |
| Subsided water body | 590.81 | 16.48 | 0 | 0 | -590.81 | -100.00 |
| Total | 3586.06 | 100 | 6524.69 | 100 | 2938.63 | 81.95 |

withdrawal of industrial and mining land in the mining subsidence area. As a result, cultivated land and land for villages and industrial and mining activities has gradually decreased in the study area and their ecological service values have declined continuously. After ecological restoration, the subsided water bodies were transformed into lake wetlands. Of 17 individual ecosystem service functions, atmospheric regulation, water conservation, entertainment, waste treatment, food production and biodiversity maintenance show an upward trend mainly because the area of water bodies increased substantially during coal mining and land reclamation in subsidence areas.

## Change of ecological storage state

Table 6 shows that ecological storage capacities were valued at RMB 6054 yuan/hm$^2$ in 2008 and RMB 12289 yuan/hm$^2$ in 2017 with an annual conversion rate of 6.15%. The results show that the entire ecosystem was developing positively from 2008 to 2017. Active ecological storage transformation mainly benefited from land reclamation and ecological restoration projects in the mining area. Through land reclamation, landscape patterns and the ecological environment in the mining subsidence area changed; land damaged by mining was reclaimed into cultivated land, woodland and grassland; and coal mining subsidence (ponding) areas were reclaimed into artificial wetlands, fish ponds, tourism and sightseeing land, etc., which has greatly improved regional landscape diversity and certainly promoted regional ecological functions. However, landscape planning has greatly expanded water bodies in the study area while sacrificing other land, inhibiting landscape diversity to an extent.

## Measurement of spillover effects of the subsided lake wetland on surrounding land prices

Residential land prices were sampled from Jiawang District and the area surrounding the Pan'an Lake wetland in Xuzhou, and data were collected from land price sample survey tables for Jiawang District for 2008 and 2017. A total of 47 residential land price samples for 2008 and 61 residential samples for 2017 were collected for the same evaluation base date. Since

**Table 6. Ecological storage state of the Pan'an Lake wetland before and after reclamation.**

| Statistics Year | Ecological storage state | Conversion quantity of ecological storage (2008–2017) | Conversion percent of ecological storage (2008–2017) |
|---|---|---|---|
| 2008 | 0.6754 | 0.5535 | 6.15 |
| 2017 | 1.2289 | | |

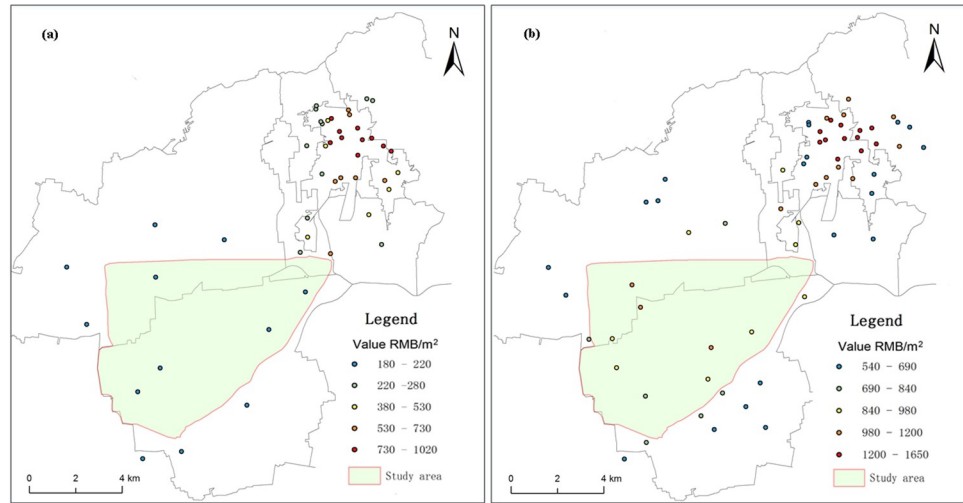

**Fig 4. Sample distribution of residential land prices in the Pan'an Lake subsidence area.** (a) Sample distribution of residential land prices in 2008; (b) Sample distribution of residential land prices in 2017.

there was less residential land around the Pan'an Lake mining subsidence lake in 2008, some industrial land price samples for the same year were selected to replace it. The distribution of land price sample points is shown in Fig 4. A normal distribution test and spatial autocorrelation analysis were performed on two sets of samples and the results show that the collected sample land prices present good overall distribution conditions and significantly positive spatial correlations and can be used for subsequent analysis.

Cross-validation was adopted to select the optimal model for Kriging spatial interpolation, and geostatistical software GS + fitting parameters were used to analyse the variation function structure [28–30]. Spatial interpolation was performed on residential land price samples for 2008 and 2017, and 15 m×15 m grid cells were determined through the graded grid division of urban land to generate the curve of residential land prices for before and after reclamation (Fig 5).

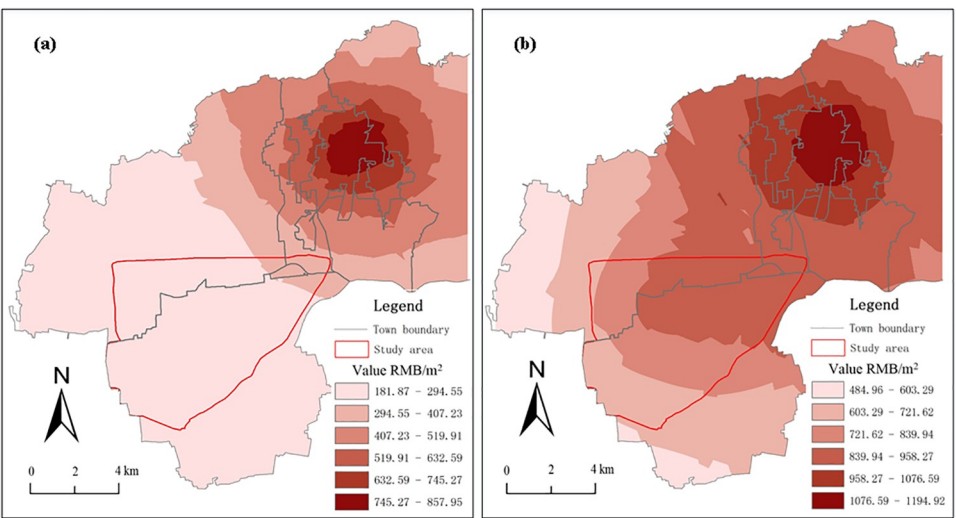

**Fig 5. Residential land prices in the Pan'an Lake subsidence area in 2008 and 2017.** (a) Residential land prices in 2008; (b) Residential land prices in 2017. The figures were constructed from Kriging interpolation results obtained from ArcGIS software.

Thus, the following formula was used to calculate the average rise in residential land prices $\Delta LP$ in the study area:

$$\Delta LP = \frac{LP_{2017} - LP_{2008}}{LP_{2008}} \times 100\% \tag{4}$$

where $LP_{2008}$ and $LP_{2017}$ are land prices in 2008 and 2017, respectively. The average rise in residential land prices in the study area is recorded as 176.81% for 2008–2017.

Land price grid data for 2017 were subtracted from land price curve data for 2008 and then from the average rise in land prices for the study area. Then, areas with values greater than 0 around the Pan'an Lake artificial wetland were set as the range within which land reclamation has affected surrounding residential land prices (Fig 6). Fig 6 shows that residential land prices are rising in the centre of the study area and significantly higher than evaluated land prices for the study area. This shows that land reclamation and ecological restoration projects in the Pan'an Lake wetland have significantly impacted land prices in and around the study area. Namely, spillover effects of land price appreciation have occurred.

A spatial overlay analysis was performed according to a planning drawing of the Pan'an Lake wetland park and the affected area in Jiawang District, and the planned residential land area in the study area was measured at 412.65 $hm^2$ as shown in Fig 6. According to the spillover effect measurement model (Eq (3)), spillover effects of the Pan'an Lake wetland on residential land in the study area amount to RMB 805,422,100 yuan according to ArcGIS 10.2 software Raster calculations.

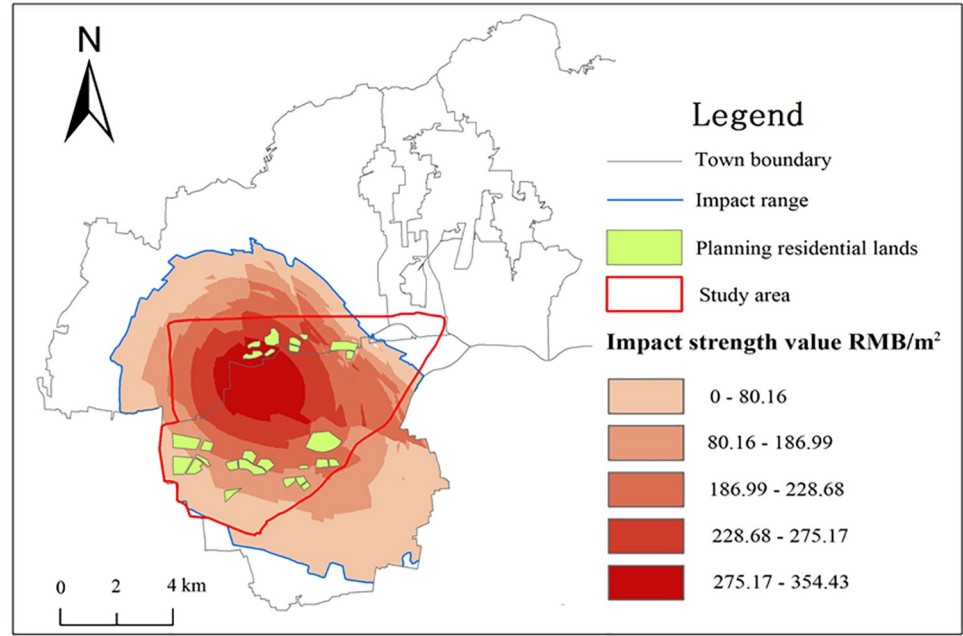

**Fig 6. Impact and range of spillover effects on residential land prices in the Pan'an Lake wetland.** The figure was drawn via ArcGIS software.

## Discussion

### Changes in the ecological environment before and after subsidence area rectification

In this paper, the ecological service value and ecological storage index were used to reflect changes in ecological conditions in the study area before and after reclamation. The results show that the ecological service value and ecological storage capacity of the Pan'an Lake wetland changed significantly before and after reclamation. From 2008 to 2017, ecosystem service functions of the Pan'an Lake wetland area were enhanced by 81.95%, and ecological storage capacities increased roughly 1.8-fold from RMB 6,754 yuan/hm$^2$ in 2008 to RMB 12,289 yuan/hm$^2$ in 2017. Throughout the life cycle, mining activities negatively impacted the ecological environment of the mining area, including production, transportation and processing activities [31, 32]. Jiawang District, a resource-exhausted city in 2011, had been a coal producing city for a century, cumulatively generating 360 million tons of raw coal. Long-term, large-scale coal mining has destroyed surrounding cultivated land; caused ground subsidence, ponding and soil and water pollution and had significantly negative effects on the ecological environment of the mining area [33–38]. From 2010, Jiawang District began to carry out land reclamation and ecological restoration projects in the coal mining area. As China's first wetland park developed through the land reclamation and ecological restoration of a coal mine subsidence area, the Pan'an Lake wetland has been constructed following a four-pronged model integrating "basic farmland rectification, coal mining subsidence reclamation, ecological environment restoration and wetland landscape development", and the constructed wetland covers roughly 10 square kilometres [39, 40]. Through land reclamation and ecological restoration, landscape patterns and ecological environments in the mining subsidence area have changed, and ecological functions such as those of atmospheric regulation, water conservation, entertainment, waste treatment, food production and biodiversity protection are improving [18]. Some studies have shown that the landscape patterns of coal mining subsidence areas change through land reclamation and management. As a result, landscape diversity is greatly increased, and the ecological quality of landscapes is gradually improved [41, 42]. Our evaluation results are largely consistent with actual conditions, indicating the effectiveness of the proposed evaluation method.

### Spillover effects of ecological rehabilitation on land prices in the mining subsidence area

In this study, a land price spillover effect model was constructed to characterize the impacts of ecological restoration on land prices in reclaimed mining areas. We applied Kriging spatial interpolation to continuous land price surfaces on the basis of land price samples obtained before and after ecological restoration on the same reference day. The average increase in land prices in the area was removed and a contour analysis was carried out to analyse changes in regional land prices before and after the ecological restoration of the studied coal mining subsided lake. As the price of residential land is susceptible to improvements in the ecological environment, the sample points of residential land prices before and after reclamation were selected for calculation and analysis. The results show that surrounding land price increases since the ecological restoration of the Pan'an Lake wetland have been significantly higher than the average rise in land prices in this region and that land prices increase toward the wetland area, showing that land reclamation and ecological restoration in the mining subsidence area have had spillover effects of increasing land value. The spillover value of Pan'an Lake wetland to residential land in the study area was calculated as RMB 805,422,100 yuan. These results

indicate that the spillover effect model of land prices constructed in this study can be used to analyse spillover effects of the ecological restoration of mining areas in the North China Plain.

## Uncertainty analysis of research results

For this study we adopted the value evaluation model of ecosystem services proposed by Costanza et al. [24] and corrected corresponding ecological service value equivalents based on actual conditions in the mining area. We then calculated the ecological service value and its changes in the study area. The evaluation results for this model may be less accurate for certain regions. However, numerous studies have proved that the method can be applied to evaluate ecosystem services in many regions of China when the equivalent correction coefficient of ecological service value developed by Xie is adopted [43–45]. In addition, because many factors may affect land prices, e.g., location, supply and demand, housing prices, ecological environments, etc. [46–48], the average growth in land prices in the study region was only used to eliminate the impacts of these factors, which may have introduced a certain degree of uncertainty. Moreover, upon eliminating the natural increase in artificial wetland prices for the mining subsidence lake, the effect of the artificial wetland on land prices was not considered, potentially causing us to underestimate the value of spillover effects. Due to the complexities and specificities of research issues, these uncertainties and limitations will be further corrected for and improved in future studies.

## Conclusions

This work addressed assessment theories and methods of ecological regimes and spatial spillover effects based on remote sensing, GIS raster calculations and geostatistical methods and analysed ecosystem changes and spatial spillover effects on land prices for a typical reclaimed mining subsided lake in Xuzhou, China. Our main conclusions are as follows:

1. After land reclamation and management, ecosystem service value and ecological storage capacity in the study area changed significantly, and the entire ecosystem has been actively and positively developing with good overall conditions. From 2008 to 2017, ecosystem service functions were enhanced by 81.95% in the constructed Pan'an Lake wetland, and ecological storage capacity increased from RMB 6,754 yuan/hm$^2$ in 2008 to RMB 12,289 yuan/hm$^2$ in 2017 at an annual conversion rate of 6.15%.

2. Transforming mining subsided lakes into artificial wetland parks has positive spillover effects on land prices where land prices increase with proximity to artificial wetland parks. The maximum value of spillover effects of residential land prices in the study area is RMB 354.43 yuan/m$^2$, the average impact value is RMB 195.18 yuan/m$^2$, and the total spillover value of planned residential land in the study area is RMB 805,422,100 yuan. These results can be used to analyse spillover effects of ecological environment restoration in reclamation areas and the economic benefits of reclamation in other mining subsidence areas.

3. The presented research methods and results can guide future studies on land reclamation planning, ecological restoration and value performance in mining subsidence areas of the North China Plain. However, due to particularities of our research subject and complexities of the issues explored, this work also presents limitations to address and remedy in future studies.

## Supporting information

**S1 File.**
(XLSX)

## Acknowledgments

We are very grateful to Jun Liang and Yanhui Zhao for their assistance with data on residential land prices and remote sensing images, Yiyan Zhang for assistance with statistical analyses. We also thank the editor and anonymous reviewers for their insightful comments on the manuscript.

## Author Contributions

**Conceptualization:** Jiaxing Xu.

**Data curation:** Jiaxing Xu, Pengcheng Yin.

**Formal analysis:** Jiaxing Xu, Hua Zhao.

**Funding acquisition:** Jiaxing Xu, Wenmin Hu.

**Investigation:** Lili Fu.

**Methodology:** Pengcheng Yin.

**Software:** Jiaxing Xu.

**Supervision:** Wenmin Hu, Lili Fu, Hua Zhao.

**Validation:** Jiaxing Xu.

**Visualization:** Jiaxing Xu.

**Writing – original draft:** Jiaxing Xu, Wenmin Hu, Hua Zhao.

**Writing – review & editing:** Jiaxing Xu, Wenmin Hu.

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
