## [Decision Letter · Decision Letter 0]

15 Apr 2020

PONE-D-20-04119

Assessing the Ecologic Regime and its Spatial Spillover Effect of a Reclaimed Mining Subsided Lake: A Case Study of the Pan'an Lake Wetland in Xuzhou

PLOS ONE

Dear Dr. Hu,

Thank you for submitting your manuscript to PLOS ONE. After careful consideration, we feel that it has merit but does not fully meet PLOS ONE’s publication criteria as it currently stands. Therefore, we invite you to submit a revised version of the manuscript that addresses the points raised during the review process.

We would appreciate receiving your revised manuscript by May 30 2020 11:59PM. To enhance the reproducibility of your results, we recommend that if applicable you deposit your laboratory protocols in protocols.io, where a protocol can be assigned its own identifier (DOI) such that it can be cited independently in the future. For instructions see: http://journals.plos.org/plosone/s/submission-guidelines#loc-laboratory-protocols

We look forward to receiving your revised manuscript.

Kind regards,

Bing Xue, Ph.D.

Academic Editor

PLOS ONE

Journal Requirements:

2. We note that Figures 1,3,5 and 6 in your submission contain map images which may be copyrighted. All PLOS content is published under the Creative Commons Attribution License (CC BY 4.0), which means that the manuscript, images, and Supporting Information files will be freely available online, and any third party is permitted to access, download, copy, distribute, and use these materials in any way, even commercially, with proper attribution. For these reasons, we cannot publish previously copyrighted maps or satellite images created using proprietary data, such as Google software (Google Maps, Street View, and Earth). For more information, see our copyright guidelines: http://journals.plos.org/plosone/s/licenses-and-copyright.

a).    You may seek permission from the original copyright holder of Figure(s) [#] to publish the content specifically under the CC BY 4.0 license.

b).    If you are unable to obtain permission from the original copyright holder to publish these figures under the CC BY 4.0 license or if the copyright holder’s requirements are incompatible with the CC BY 4.0 license, please either i) remove the figure or ii) supply a replacement figure that complies with the CC BY 4.0 license. Please check copyright information on all replacement figures and update the figure caption with source information. If applicable, please specify in the figure caption text when a figure is similar but not identical to the original image and is therefore for illustrative purposes only.

3. Thank you for stating the following in the Financial Disclosure section:"This work is jointly supported by the National Natural Science Foundation of China (41401610, 41601500) and National Key Research and Development Program of China (2017YFC0804401). The funders had no role in study design, data collection and analysis, decision to publish, or preparation of the manuscript. "

Thank you for stating the following in the Competing Interests section:"The authors have declared that no competing interests exist."

We note that one or more of the authors are employed by a commercial company:"Jiangsu Zenith Geo-Informatics Co., Ltd"

b). Please also provide an updated Competing Interests Statement declaring this commercial affiliation along with any other relevant declarations relating to employment, consultancy, patents, products in development, or marketed products, etc. 

4. Please amend your data availability statement to declare where data was collected from, and how others can access it. Please also ensure that all data sources are clearly named and referenced in the manuscript.

Reviewers' comments:

Reviewer's Responses to Questions

**Comments to the Author**

1. Is the manuscript technically sound, and do the data support the conclusions?

Reviewer #1: Yes

Reviewer #2: Yes

Reviewer #3: Partly

Reviewer #4: Partly

2. Has the statistical analysis been performed appropriately and rigorously? 

Reviewer #1: Yes

Reviewer #2: Yes

Reviewer #3: No

Reviewer #4: I Don't Know

3. Have the authors made all data underlying the findings in their manuscript fully available?

Reviewer #1: Yes

Reviewer #2: Yes

Reviewer #3: No

Reviewer #4: No

4. Is the manuscript presented in an intelligible fashion and written in standard English?

Reviewer #1: Yes

Reviewer #2: No

Reviewer #3: No

Reviewer #4: Yes

5. Review Comments to the Author

Reviewer #1: The MANUSCRIPT is discussing a good topic but the technical methodology needs some exploration

I have some criticisms, which are mainly summarized in the following points:

- The innovation of the paper seems quite limited; the methods applied are widely known. Further, the methods are only partially appropriate for the data analysis; for instance, some of the data represent proportions/percentages and this does not seem to be accounted for in the analysis.

- The models and methods are poorly detailed and justified, some assumptions made by the authors seem clearly violated; the analyses would be hardly replicable.

The validation of the statistical analysis appears very poor.

For these reasons, the submitted manuscript does appear to be suitable for publication in PLOS ONE after some responding to the previous comments.

Reviewer #2: This manuscript selected a representative reclaimed wetland park in Xuzhou city to assess the ecologic regime change and spillover effect, the results demonstrated that the ecosystem service value and ecological storage capacity have changed after the subsided waterbody changed to wetland, which will help improve the land price of surrounding areas.

I am excited to see an example study of land subsidence reclamation. However, I have several concerns that need carefully review and improvement before publication.

Writing and grammar:

The manuscript needs a thoroughly writing editing.

1. The writing of this manuscript is hard to understand, to name a few,

Line 23: “has been attempted in China”, and Line 70 “an attempt to”

Line 27: “wetland derived from”

Line 111:”With …”

Line 131: “had been completed”

The wording and grammar need more carful checking.

2. I found the manuscript maybe was from a dissertation, for there are two words of “dissertation” showed in Line 156 and 387, which should be revised.

3. There are some spelling mistakes in the manuscript, such as Line 238 “landscape”, Line 330 “calculation”.

Name unifying:

4. In Abstract section, “eastern China” was used to locate the study area, while in Line 59, it turned into “North China”, please unify the diction. Generally, we will use north China or North China Plain to indicate location of the high water table mining area.

5. In Table 3, you used “Urban/ Industry/Mining area”, while you named it “urban/rural construction land and industrial/mining construction land” in Line 144.

6. In Table 4, “Natural Waterbody”, while you used “general water bodies” in Line 143.

Figures and Tables:

7. Figure 1. There is need to put a compass in the figure.

8. Figure 1. From my view, the redline areas is not just a wetland, it is the scope of a mining region.

9. Figure 2. Upper right, what is the meaning of “Planned land”?

Reviewer #3: This paper followed the remote sensing, GIS raster calculation and geostatistical methods to assess the ecosystem changes and their spatial spillover effect in mining subsidence area of Pan'an Lake Wetland, China. Here are some comments or suggestions for improving the current version.

1. The Introduction section of the thesis is insufficient to review existing related research, for example, lack of review and comparison of the advantages and disadvantages of related calculations and research methods. In addition, the review of the literature should be the author's own summary, rather than citing other literature. Finally, the conclusions of the literature review are not convincing enough for the reasons why the authors should carry out this research.

2. From the data, the author collected 47 residential land price samples in 2008 and 65 residential land price samples in 2004. Compared with the scope of the study area, the amount collected is too small, which directly affects the accuracy of spatial interpolation results.

3. As we all know, in recent years China's housing and land prices have shown an overall rapid upward trend. Even if it is not near the case lake, housing and land prices will still show rising results. Only by accurately crediting the price increase to the ecologic regime and the ecological spillover effect can readers be convinced.

4. What are the implications of the case studies in this paper for other regions of China or for other countries? The author needs further explanation.

5. The paper lacks the refinement of the research contribution and the comparative discussion with the existing research.

6. In addition, the display quality of the figures in the paper is not enough. There are some errors in spelling and English writing.

This paper is not suitable for publication in its current form.

Reviewer #4: The paper suggests interesting methodologies to account for ecological value and spillover effects of restoration projects, specifically subsided lakes turned into artificial wetland. However, major improvements are needed to make it suitable for publication. In particular, the methods are poorly described, so undermining the capability to appreciate the results. The results are not properly emphasized and commented (i.e. why ecological and spillover considerations are important, model applicability in other scenarios, etc). I suggest a major revision for the ms and attach some notes that could help to improve the paper

6. PLOS authors have the option to publish the peer review history of their article (what does this mean?). If published, this will include your full peer review and any attached files.

Reviewer #1: Yes: Mohamed A.E. AbdelRahman

Reviewer #2: No

Reviewer #3: No

Reviewer #4: No

---

## [Author Response · Author response to Decision Letter 0]

14 Jul 2020

Responses to editor’s and reviewers’ comments

Responses to editor’s comments:

RE: Thank you for your information. We have revised the manuscript as much as possible to meets PLOS ONE's style requirements. 

2. We note that Figures 1,3,5 and 6 in your submission contain map images which may be copyrighted. All PLOS content is published under the Creative Commons Attribution License (CC BY 4.0), which means that the manuscript, images, and Supporting Information files will be freely available online, and any third party is permitted to access, download, copy, distribute, and use these materials in any way, even commercially, with proper attribution. For these reasons, we cannot publish previously copyrighted maps or satellite images created using proprietary data, such as Google software (Google Maps, Street View, and Earth). For more information, see our copyright guidelines: http://journals.plos.org/plosone/s/licenses-and-copyright.

a). You may seek permission from the original copyright holder of Figure(s) [#] to publish the content specifically under the CC BY 4.0 license.

b). If you are unable to obtain permission from the original copyright holder to publish these figures under the CC BY 4.0 license or if the copyright holder’s requirements are incompatible with the CC BY 4.0 license, please either i) remove the figure or ii) supply a replacement figure that complies with the CC BY 4.0 license. Please check copyright information on all replacement figures and update the figure caption with source information. If applicable, please specify in the figure caption text when a figure is similar but not identical to the original image and is therefore for illustrative purposes only.

RE: There are no copyright restrictions on these four figures. We added the details of the figures in the revised manuscript. 

(1) In Figure 1, the remote sensing base map is replaced by the OpenStreetMap from USGS National Map viewer. The map is created using ArcGIS software and it is subject to no copyright restrictions. (Lines 140-143)

(2) In Figure 3, Figure 3(a) and 3(b) are based on the results of remote sensing images interpretations conducted using ENVI software. Remote sensing images were described in the section of “Data sources and processing” of the revised manuscript. So these figures have no copyright restrictions. (Lines 259-262)

 (3) In Figure 5, Figure 5(a) and 5(b) are the results of Kriging interpolation using ArcGIS software. So they have no copyright restrictions. (Lines 326-328)

(4) Figure 6 was drawn via ArcGIS software. So this figure has no copyright restrictions. (Lines 343-344)

3. Thank you for stating the following in the Financial Disclosure section:"This work is jointly supported by the National Natural Science Foundation of China (41401610, 41601500) and National Key Research and Development Program of China (2017YFC0804401). The funders had no role in study design, data collection and analysis, decision to publish, or preparation of the manuscript. "

Thank you for stating the following in the Competing Interests section:"The authors have declared that no competing interests exist."

We note that one or more of the authors are employed by a commercial company:"Jiangsu Zenith Geo-Informatics Co., Ltd"

b). Please also provide an updated Competing Interests Statement declaring this commercial affiliation along with any other relevant declarations relating to employment, consultancy, patents, products in development, or marketed products, etc. 

RE: Thank you for your information. We updated Funding Statement and Competing Interests Statement in cover letter. And we also added the following information in the Funding Statement and Competing Interests Statement in the revised manuscript submission.

In Funding Statement, we added the following information. 

“The commercial company ’Jiangsu Zenith Geo-Informatics Co., Ltd’ provided support in the form of salaries for author Lili Fu, but did not have any additional role in the study design, data collection and analysis, decision to publish, or preparation of the manuscript.”

In Competing Interests Statement, we added the following information.

“The commercial company ’Jiangsu Zenith Geo-Informatics Co., Ltd’ does not provide experimental funds and will not use the relevant information of the paper to develop products and apply for patents, and will not mind appearing as an author unit. This does not alter our adherence to PLOS ONE policies on sharing data and materials”.

We updated Funding Statement and Competing Interests Statement as following: 

Funding Statement: This work is jointly supported by the National Natural Science Foundation of China (41401610, 41601500 and 51874278) and National Key Research and Development Program of China (2017YFC0804401) and the Independent Research Project of State Key Laboratory of Coal Resources and Safe Mining (SKLCRSM2020X04). The funders had no role in study design, data collection and analysis, decision to publish, or preparation of the manuscript. The commercial company ’Jiangsu Zenith Geo-Informatics Co., Ltd’ provided support in the form of salaries for the author Lili Fu, but did not have any additional role in the study design, data collection and analysis, decision to publish, or preparation of the manuscript.”

Competing Interests Statement: The authors have declared that no competing interests exist. The commercial company ’Jiangsu Zenith Geo-Informatics Co., Ltd’ does not provide experimental funds and will not use the relevant information of the paper to develop products and apply for patents, and will not mind appearing as an author unit. This does not alter our adherence to PLOS ONE policies on sharing data and materials.

4. Please amend your data availability statement to declare where data was collected from, and how others can access it. Please also ensure that all data sources are clearly named and referenced in the manuscript.

RE: Thank you for your information. We explain the source of the data as following. 

1) The remote sensing data used in the study were obtained through purchase. SPOT5 image acquired on October 16, 2008 can be inquired through the website (http://eds.ceode.ac.cn) and China’s GF2 images acquired on October 19 and 29, 2017 can be inquired through the website for China Resource Satellite Application Center (http://36.112.130.153:7777/DSSPlatform/productSearch.html).

2) Land price samples data was from Jiawang land price dynamic monitoring sample survey, which were the result of our long-term monitoring and evaluation of land prices. The base date of evaluation of the land price samples were 31 December 2008 and 31 December 2017, respectively. Price samples were collected in the study area for 47 residential lands in 2008 and 65 residential lands in 2017, for which spatial interpolation was performed using GIS and geostatistical methods.

We have provided the original data, which has been uploaded as an attachment.

We updated Data availability statement as following: 

Data availability statement: SPOT5 image acquired on October 16, 2008 can be inquired through the website (http://eds.ceode.ac.cn) and China’s GF2 images acquired on October 19 and 29, 2017 can be inquired through the website for China Resource Satellite Application Center (http://36.112.130.153:7777/DSSPlatform/productSearch.html).The data of land price samples is within the manuscript and its Supporting Information files. 

Additional Editor Comments:

1. Thank you for providing the data use agreement with China Resource Satellite Application Center.

We note this data use agreement states Party B shall not copy, transfer or lend it without the written permission of Party A. Additionally, PLOS ONE's utilizes the CC BY 4.0 license (https://creativecommons.org/licenses/by/4.0/) which means that all material on our website is freely available online, and any third party is permitted to access, download, copy, distribute, and use these materials in any way, even commercially, with proper attribution. It is not clear in the data use agreement if it allows the data to be used for commercial purposes or only for scientific research.

Therefore, we require specific consent from the copyright holder to publish these images in PLOS ONE, under the CC BY 4.0 license. To seek permission from the copyright owner to publish your map figures under the specific Creative Commons Attribution License (CCAL), CC BY 4.0, please contact them with the following text and PLOS ONE Request for Permission form (http://journals.plos.org/plosone/s/file?id=7c09/content-permission-form.pdf):

“I request permission for the open-access journal PLOS ONE to publish XXX under the Creative Commons Attribution License (CCAL) CC BY 4.0 (http://creativecommons.org/licenses/by/4.0/). Please be aware that this license allows unrestricted use and distribution, even commercially, by third parties. Please reply and provide explicit written permission to publish XXX under a CC BY license.”

Please upload the granted permission to the manuscript as a Supporting Information file. In the figure caption of the copyrighted figure, please include the following text: “Republished from [ref] under a CC BY license, with permission from [name of publisher], original copyright [original copyright year].”

Please note that RightsLink permission forms often impose use restrictions that are incompatible with our CC BY 4.0 license, and we are therefore unable to accept these permissions. For this reason, we strongly recommend contacting copyright holders with the PLOS ONE Request for Permission form.

If you are unable to obtain permission from the copyright holder, please either A) remove the figure or B) supply a replacement figure that complies with the CC BY 4.0 license. Please check copyright information on all replacement figures and update the figure caption with source information.

USGS National Map Viewer (http://viewer.nationalmap.gov/viewer/)

USGS Earth Resources Observatory and Science (EROS) Center (http://eros.usgs.gov/#)

The Gateway to Astronaut Photography of Earth (https://eol.jsc.nasa.gov/)

Maps at the CIA (https://www.cia.gov/library/publications/the-world-factbook/docs/refmaps.html)

NASA Earth Observatory (http://earthobservatory.nasa.gov/)

Landsat (http://landsat.visibleearth.nasa.gov/)

Natural Earth (http://www.naturalearthdata.com/)

RE: Thank you for your information. We are very sorry that we cannot obtain permission from the copyright holder. Therefore, the remote sensing base map in Figure 1 is replaced by the OpenStreetMap from USGS National Map viewer. The map is created using ArcGIS software and it is subject to no copyright restrictions.

We modified it in Figure 1. Please see the revised manuscript for details. (Lines 140-143)

2. Please clarify in your Data sources and processing section how the land price land price sample survey tables for Jiawang District for 2008 and 2017 were obtained by the authors. Please also clarify how the satellite data was obtained by the authors.

RE: Thank you for your information. We explain as follows:

1) We explained the source of the land price samples in the ‘Data Sources and processing’ section of the revised manuscript. We have provided the original data, which has been uploaded as an attachment.

“Land price samples data was from Jiawang land price dynamic monitoring sample survey, which were the result of our long-term monitoring and evaluation of land prices. The base date of evaluation of the land price samples were 31 December 2008 and 31 December 2017, respectively. Price samples were collected in the study area for 47 residential lands in 2008 and 65 residential lands in 2017, for which spatial interpolation was performed using GIS and geostatistical methods.” (Lines 160-165)

2) The remote sensing data used in the study were obtained through purchase. SPOT5 image acquired on October 16, 2008 can be inquired through the website (http://eds.ceode.ac.cn) and China’s GF2 images acquired on October 19 and 29, 2017 can be inquired through the website for China Resource Satellite Application Center (http://36.112.130.153:7777/DSSPlatform/productSearch.html). 

In the revised manuscript, we introduce the information and source of remote sensing data in detail. (Lines 147-152)

Responses to reviewers’ comments

Reviewer #1: The MANUSCRIPT is discussing a good topic but the technical methodology needs some exploration

RE: Thanks for your positive comments. We have carefully revised the manuscript according to your comments and suggestions. 

I have some criticisms, which are mainly summarized in the following points:

- The innovation of the paper seems quite limited; the methods applied are widely known. Further, the methods are only partially appropriate for the data analysis; for instance, some of the data represent proportions/percentages and this does not seem to be accounted for in the analysis.

RE: Thanks for your suggestions. We made an explanation as following:

 (1) In this paper, firstly, we use the ecosystem service value and the ecological storage index to analyze the ecological changes in the reclaimed mining area. The emphasis is to analyze the impact of ecological restoration of reclaimed mining areas on the surrounding land price by building a model of land price spillover effect. It can clearly measure the influence range, intensity and total value of the artificial wetland of coal mining subsided lake on the surrounding land price spillover effect, and could provide a research basis for revealing the value performance of the artificial wetland of coal mining subsided lake. These are the innovation of this paper. 

The spillover effect model of land price was built based on land statistics and GIS raster calculation, which are the mature and common methods and can guarantee the accuracy of calculation. 

(2) In the process of data analysis, we analyzed some typical features of the data, so did not explain all the data features. We can view the data in the tables.

- The models and methods are poorly detailed and justified, some assumptions made by the authors seem clearly violated; the analyses would be hardly replicable.

The validation of the statistical analysis appears very poor.

For these reasons, the submitted manuscript does appear to be suitable for publication in PLOS ONE after some responding to the previous comments.

RE: Thanks for your suggestions. We made an explanation as following:

(1) We modified the expression of research method to make it clearer. Please see the Methodology section of the revised manuscript for details. (Lines 170-176、Lines 219-234)

(2) In this study, the typical mining subsidence lake was taken as the research area. The data included remote sensing images in 2008 and 2017, land price sample data before and after reclamation, which were introduced in the manuscript. 

As a typical case of reclaimed mining subsided lake, the study has certain representativeness, and the research results have theoretical and practical significance for the ecological restoration and management of mining areas with high groundwater level in the North China Plain. However, there are uncertainties in the research results. For example, when the natural increase factor in the artificial wetland price of the mining subsidence lake is eliminated, the effect of the artificial wetland on the land price has not been considered, which may make the calculated value of the spillover effect to be on the low side.

We discussed the uncertainty of the research results in the discussion section of the revised manuscript. (Lines 404-418)

Reviewer #2: This manuscript selected a representative reclaimed wetland park in Xuzhou city to assess the ecologic regime change and spillover effect, the results demonstrated that the ecosystem service value and ecological storage capacity have changed after the subsided waterbody changed to wetland, which will help improve the land price of surrounding areas.

I am excited to see an example study of land subsidence reclamation. However, I have several concerns that need carefully review and improvement before publication.

RE: Thanks for your positive comments. We have carefully revised the manuscript according to your comments and suggestions.

Writing and grammar:

The manuscript needs a thoroughly writing editing.

1. The writing of this manuscript is hard to understand, to name a few,

RE: Thank you for pointing out the mistakes in writing and grammar in the manuscript. We have carefully checked and corrected them according to your comments, which are highlighted in red. Please see the revised manuscript for details.

Line 23: “has been attempted in China”, and Line 70 “an attempt to”

RE: Sentences have now been rewritten.

“In China, mining subsided lakes are often transformed into artificial wetland parks for ecological restoration.” (Lines 29-30)

“In recent years, several mining subsided lakes in China have been transformed into artificial wetland parks according to local conditions using land reclamation techniques as a means of ecological restoration.” (Lines 74-76)

Line 27: “wetland derived from”

RE: Sentences have now been rewritten.

“This paper examines the Pan'an Lake artificial wetland in Jiawang District, Xuzhou, as a case study.” (Lines 33-34)

Line 111:”With …”

RE: Sentences have now been rewritten.

“This paper focused on the Pan'an Lake wetland in Jiawang District, Xuzhou. Ecosystem service value and the ecological storage index were used to ...” (Lines 117-120)

Line 131: “had been completed”

RE: Sentences have now been rewritten.

“A national 4A-level ecological wetland park with a water area of 9.21 km2 and an overall area of 52.89 km2 has been constructed.” (Lines 136-137)

2. I found the manuscript maybe was from a dissertation, for there are two words of “dissertation” showed in Line 156 and 387, which should be revised.

RE: These are errors of expression. 

The word “dissertation” showed in Line 156 was modified. (Lines 175) 

The word “dissertation” was replaced by the word “study”. (Line 412)

3. There are some spelling mistakes in the manuscript, such as Line 238 “landscape”, Line 330 “calculation”.

RE: Thank you for your examination and revision of the spelling mistakes in the manuscript. We have revised them in the revised paper, which are highlighted in red. 

Name unifying:

4. In Abstract section, “eastern China” was used to locate the study area, while in Line 59, it turned into “North China”, please unify the diction. Generally, we will use north China or North China Plain to indicate location of the high water table mining area.

RE: According to your suggestion, we change them to "North China Plain". (Line 28)

5. In Table 3, you used “Urban/ Industry/Mining area”, while you named it “urban/rural construction land and industrial/mining construction land” in Line 144.

RE: In the land use classification of this study, built-up land includes urban and rural built-up land, industrial and mining built-up land. 

We changed it to "built-up land" in Table 3 and explained the classification in the section of data sources and processing of the revised manuscript. (Line 158)

We also modified it in Figure 3. 

Please see the revised manuscript for details.

6. In Table 4, “Natural Waterbody”, while you used “general water bodies” in Line 143.

RE: Thank you for your information. We use "natural water body". It has been modified in the revised manuscript. (Line 157)

Figures and Tables:

7. Figure 1. There is need to put a compass in the figure.

RE: We added a compass in the Figure 1 according to your suggestion.

8. Figure 1. From my view, the redline areas is not just a wetland, it is the scope of a mining region.

RE: Yes. The redline area in Figure 1 is the study area which demarcated according to the scope of the reclamation project. At the same time, it is also the mining region of the two coal mines (Qishan mine and Quantai mine). A large number of mining subsidence lakes have been formed after coal mining.

9. Figure 2. Upper right, what is the meaning of “Planned land”?

RE: We originally intended to present land planning data in the manuscript. We used “Land planning data” instead of “planned land” in Figure 2.

Reviewer #3: This paper followed the remote sensing, GIS raster calculation and geostatistical methods to assess the ecosystem changes and their spatial spillover effect in mining subsidence area of Pan'an Lake Wetland, China. Here are some comments or suggestions for improving the current version.

RE: Thanks for your positive comments. We have carefully revised the manuscript according to your comments and suggestions.

1. The Introduction section of the thesis is insufficient to review existing related research, for example, lack of review and comparison of the advantages and disadvantages of related calculations and research methods. In addition, the review of the literature should be the author's own summary, rather than citing other literature. Finally, the conclusions of the literature review are not convincing enough for the reasons why the authors should carry out this research.

RE: Thanks for your suggestions. 

(1) In the introduction section of the revised draft, we analyzed and summarized the relevant research on the ecological restoration of the artificial wetland caused by coal mining collapse, and pointed out the shortcomings of the current research, which are the basis for this research. (Lines 82-94)

(2) The advantages and disadvantages of related research methods are compared, and the calculation method suitable for this study is proposed. (Lines 95-116)

Please see the introduction section of the revised manuscript for details.

2. From the data, the author collected 47 residential land price samples in 2008 and 65 residential land price samples in 2004. Compared with the scope of the study area, the amount collected is too small, which directly affects the accuracy of spatial interpolation results.

RE: We made an explanation as following.

The sample data of residential land price comes from the survey data of land price sample points in Jiawang District. Land price survey focuses on the dynamic monitoring of the urban land price, so the distribution characteristic of land price sample data is that urban sample points are more than suburban sample points. The study area is relatively remote from the urban area, mainly for mining area and supporting facilities, and there are few residential areas. Therefore, the amount of residential land price samples is small in the study area, which reflects the actual situation of land price in the study area. 

Based on the correlation and variability of variables, the Kriging spatial interpolation method performs an unbiased and optimal estimation of the values of regionalized variables in a finite region. It has obvious advantages: in the process of data gridding, the spatially related properties of the described objects are considered, so that the estimated results are more scientific and closer to the actual situation. The method has certain advantages for data with small density and uneven distribution.

In this study, the spatial distribution of land price is obtained by interpolation of land price sample points, which is mainly used to analyze the impact of ecological restoration in mining area on the surrounding land price. Kriging spatial interpolation method is used to interpolate the sampling points, which can meet the requirements of the study.

3. As we all know, in recent years China's housing and land prices have shown an overall rapid upward trend. Even if it is not near the case lake, housing and land prices will still show rising results. Only by accurately crediting the price increase to the ecologic regime and the ecological spillover effect can readers be convinced.

RE: Yes. Land prices are jointly affected by various factors. Only by excluding the influence of other factors can we determine that the change of land price is caused by the ecological restoration of the mining area. 

In this study, a land price spillover model was constructed to characterize the impact of ecological restoration on land price in reclaimed mining areas. First of all, we analyzed the influencing factors of land price in the study area. Affected by the housing price situation in China, the housing price and land price in Jiawang District are also growing rapidly. In addition, except for the ecological restoration project, there are no other major project implementation and other factors affecting the land price in the study area. Therefore, it can be considered as the impact of ecological restoration project on land price as long as the average increase value of land price in the region is removed.

According to the above ideas, this paper puts forward the spillover effect model of a reclaimed mining subsided lake on the surrounding land price as follows:

where represents the average increase value of land price in the study area during the study period, and we subtracted this value from the model. 

Therefore, the results calculated by the land price spillover effect model are relatively reasonable.

4. What are the implications of the case studies in this paper for other regions of China or for other countries? The author needs further explanation.

RE: This paper proposes a spillover effect model of land price based on land statistics and GIS raster calculation, which can clearly measure the influence range, intensity and total value of the artificial wetland of coal mining subsidence lake on the surrounding land price spillover effect, and provide a research basis for revealing the value performance of the artificial wetland of coal mining subsidence lake.

The paper only takes Pan'an Lake wetland as an example to make an empirical analysis. The case study has certain representativeness, and the research results have theoretical and practical significance for the ecological restoration and management of mining areas with high groundwater level in the North China Plain. It can also be used for reference for the evaluation of ecological restoration effect of similar geographical environment in foreign countries.

We supplement the application of the research results in the discussion and conclusion section of the revised manuscript. (Lines 399-401 and Lines 438-440 )

5. The paper lacks the refinement of the research contribution and the comparative discussion with the existing research.

RE: This paper focuses on the analysis of the change of ecosystem status and its spatial spillover effect on land price after the coal mining subsided lake becomes an artificial wetland park. According to your suggestions, we have refined the research results and compared with the existing research in the discussion and conclusion section of the revised manuscript. 

Please see the discussion and conclusion section of the revised manuscript for details. (Lines 355-356, Lines 377-380, Lines 385-391, Lines 421-424 and Lines 431-433)

6. In addition, the display quality of the figures in the paper is not enough. There are some errors in spelling and English writing.

RE: (1) We revised Figure 1, Figure 2, and Figure 3. All the high-quality figures have been uploaded as attachments. 

(2) According to the reviewer's suggestion, we have modified the spelling and grammatical errors of the whole paper, which are highlighted in red. 

(3) We have employed a professional language service ‘American Journal Experts (AJE)’ to revise the manuscript. 

Please see the revised manuscript for details.

Reviewer #4: The paper suggests interesting methodologies to account for ecological value and spillover effects of restoration projects, specifically subsided lakes turned into artificial wetland. However, major improvements are needed to make it suitable for publication. In particular, the methods are poorly described, so undermining the capability to appreciate the results. The results are not properly emphasized and commented (i.e. why ecological and spillover considerations are important, model applicability in other scenarios, etc). I suggest a major revision for the ms and attach some notes that could help to improve the paper.

RE: Thank you very much for your constructive comments on our manuscript. According to your suggestion, we have carefully revised the manuscript as following.

(1) The expression of the research method was modified to make it clearer. Please see the Methodology section of the revised manuscript for details. (Lines 170-176 and Lines 219-234)

(2) As we know that land reclamation is an important way of ecological restoration and management in mining areas, which was carried out earlier in eastern mining areas of China. In recent decades, what are the benefits of ecological restoration in the reclamation area? And which index is chosen to evaluate the benefit of ecological restoration? It has become the focus of many scholars. 

The land reclamation benefits of mining areas are mostly measured based on economic benefits, such as increases in cultivated land and increases and decreases in built-up land, while less attention is paid to ecological environmental effects after reclamation. On one hand, the ecological restoration of a reclaimed mining area involves a long-term restoration process, and due to the influence of reclamation measures, coal mining and other factors, ecological restoration is often not supported with long-term quantitative monitoring. On the other hand, as public resources, ecological environments have obvious spatial spillover effects. However, current research on the ecological environments of reclaimed mining areas mainly focuses on ecological problems experienced within such areas based on evaluations of ecosystem services, water pollution and nutrition status monitoring, landscape ecological restoration, etc. [17-19] while few studies examine impacts of the ecological restoration of a mining area on the surrounding environment such as spillover effects of the improvement of an ecological environment on surrounding land prices.

The importance of the research is explained and modified in the introduction section of the revised manuscript. (Lines 63-94)

(3) The case study has certain representativeness, and the research results have theoretical and practical significance for the ecological restoration and management of mining areas with high groundwater level in the North China Plain. It can also be used for reference for the evaluation of ecological restoration effect of similar geographical environment in foreign countries.

We supplement the application of the research results in the discussion and conclusion section of the revised manuscript. (Lines 399-401 and Lines 438-440 )

(4) We revised the manuscript, especially the spelling and grammar. Please see the revised manuscript for details.

---

## [Decision Letter · Decision Letter 1]

13 Aug 2020

Assessing the ecological regime and spatial spillover effects of a reclaimed mining subsided lake: a case study of the Pan'an Lake wetland in Xuzhou

PONE-D-20-04119R1

Dear Dr. Hu,

We’re pleased to inform you that your manuscript has been judged scientifically suitable for publication and will be formally accepted for publication once it meets all outstanding technical requirements.

Kind regards,

Bing Xue, Ph.D.

Academic Editor

PLOS ONE

Additional Editor Comments (optional):

Reviewers' comments:

Reviewer's Responses to Questions

**Comments to the Author**

1. If the authors have adequately addressed your comments raised in a previous round of review and you feel that this manuscript is now acceptable for publication, you may indicate that here to bypass the “Comments to the Author” section, enter your conflict of interest statement in the “Confidential to Editor” section, and submit your "Accept" recommendation.

Reviewer #2: All comments have been addressed

Reviewer #3: All comments have been addressed

2. Is the manuscript technically sound, and do the data support the conclusions?

Reviewer #2: Partly

Reviewer #3: Yes

3. Has the statistical analysis been performed appropriately and rigorously? 

Reviewer #2: Yes

Reviewer #3: Yes

4. Have the authors made all data underlying the findings in their manuscript fully available?

Reviewer #2: Yes

Reviewer #3: Yes

5. Is the manuscript presented in an intelligible fashion and written in standard English?

Reviewer #2: No

Reviewer #3: Yes

6. Review Comments to the Author

Reviewer #2: I am happy to see that all the questions are addressed by the authors. The manuscript can be accepted for publication at this stage.

Reviewer #3: The suggested recommendations have been correctly and almost accurately addressed.

7. PLOS authors have the option to publish the peer review history of their article (what does this mean?). If published, this will include your full peer review and any attached files.

Reviewer #2: No

Reviewer #3: No

---

## [Editor Report · Acceptance letter]

17 Aug 2020

PONE-D-20-04119R1 

Assessing the ecological regime and spatial spillover effects of a reclaimed mining subsided lake: a case study of the Pan'an Lake wetland in Xuzhou 

Dear Dr. Hu:

I'm pleased to inform you that your manuscript has been deemed suitable for publication in PLOS ONE. Congratulations! Your manuscript is now with our production department. 

Kind regards, 

on behalf of

Professor Bing Xue 

Academic Editor

PLOS ONE